# Knowledge, Attitude, and Practices on Tomato Leaf Miner, *Tuta absoluta* on Tomato and Potential Demand for Integrated Pest Management among Smallholder Farmers in Kenya and Uganda

**Fridah Chepchirchir [1,2], Beatrice W. Muriithi [2,*], Jackson Langat [1], Samira A. Mohamed [2], Shepard Ndlela [2] and Fathiya M. Khamis [2]**

1 Department of Agricultural and Agribusiness Management, Egerton University, Njoro P.O. Box 536-20115, Kenya; chepchirchir.1465818@student.egerton.ac.ke (F.C.); jlangat@egerton.ac.ke (J.L.)
2 International Centre of Insect Physiology and Ecology (icipe), Nairobi P.O. Box 30772-00100, Kenya; sfaris@icipe.org (S.A.M.); sndlela@icipe.org (S.N.); fkhamis@icipe.org (F.M.K.)
* Correspondence: bmuriithi@icipe.org

**Abstract:** Agricultural growth and food security are a priority in many developing countries. This has led to increased attention to effective pest management. Integrated Pest Management (IPM) strategy is a sustainable and recommended alternative to the use of synthetic pesticides in the management of tomato pests, with *Tuta absoluta* being the major one. This study seeks to assess the awareness, attitude, and control practices on *T. absoluta* and examine the potential adoption of a proposed IPM strategy for the management of a pest using a randomly selected sample of 316 and 345 tomato growing households in Kenya and Uganda, respectively. The study findings indicate that *T. absoluta* is the major pest affecting tomato production, with most farmers using synthetic pesticides to manage it. Furthermore, we find a significant proportion of the survey respondents willing to adopt the IPM strategy. The probability of adopting the strategy was positively related to a farmer being male, residing near a source of inputs, accessing training, and possessing good knowledge, attitude, and practices towards the use of non-pesticides strategies. Thus, training, promotion, and awareness creation of the *T. absoluta* IPM are recommended for the sustainable management of the pest in tomato production.

**Keywords:** Integrated Pest Management; *Tuta absoluta*; *ex-ante* adoption; bayesian analysis; Kenya; Uganda

## 1. Introduction

Tomato (*Solanum lycopersicum* L.) is one of the most popular vegetables in sub-Saharan Africa owing to its nutritional value and functions as both a food and cash crop [1]. Furthermore, tomato offers a reliable source of employment and income generation to small- and medium-scale growers. However, the current production is below the potential level. For instance, in Kenya, the current production is 283,000 tonnes per hectare while that of Uganda contributes 40,124 tonnes per hectare [2], in contrast to the estimated potential of 300,000 tonnes per hectare. The gap is attributed to a myriad of challenges, key among them being insect pests and diseases. Currently, the tomato leaf miner, *Tuta absoluta*, is the major insect pest affecting tomato production in East Africa. It infests the leaves, stems, and fruits [3], causing between 80–100% loss in yield, both in protected and native fields if left uncontrolled [4]. The high losses in East Africa are attributed to the warm climatic conditions [5]. Despite the heavy use of synthetic pesticides to control this pest, the production loss attributed to it remains high [6], besides the health and environmental risks and resistance associated with the use (and misuse) of chemical pesticides [7].

An alternative crop protection paradigm, such as Integrated Pest Management (IPM) approaches or strategies have been recommended by scientists, policymakers, and international development agencies to hazardous synthetic insecticides [8–10]. Although widely promoted and used in developed countries, the adoption of IPM in developing countries including Africa is generally limited. The definitions of IPM are numerous, however, all of them involve the coordinated integration of multiple complementary practices to manage the pest in a safe, cost-effective, and environmentally friendly manner [8,11,12]. The positive economic, environmental, and social impacts of IPM are evident in Africa. A good example is the use of fruit fly suppressing IPM in mango production [13–15]. These studies demonstrated decreased losses associated with the pest and thus higher yield and income, as well as reduced pesticide-related risks to human health and the environment. Previous studies on IPM used in the management of tomato-infesting pests and diseases have equally demonstrated positive impacts. For example, a study on integrated pests and disease management in tomato production in India found that IPM increased tomato yield by 46%, reduced the cost of cultivation by 21%, and increased net returns by 19% [16].

In Africa, the International Centre of Insect Physiology and Ecology (icipe) in collaboration with development partners seeks to introduce an IPM package for suppressing the tomato infesting *T. absoluta*. Although IPM practices exhibit potential economic and environmental benefits to tomato growers and the horticultural subsector in the SSA region, wide-scale commercialization and adoption of the technology will depend on farmers' pre-conceived perceptions, preferences, and their acceptance of the new technology. This study, therefore, conducted before disseminating the IPM strategy, seek to assess the knowledge, attitude, and practices of tomato growing households with regard to *T. absoluta*, and determine the potential demand for the strategy for sustainable management of the pest, using data obtained from selected tomato producing regions in Kenya and Uganda.

This study contributes to the existing literature on *ex-ante* adoption of agricultural innovations in two folds. First, this is the first study to estimate potential demand for the proposed icipe's IPM approaches for the management of the *T. absoluta*. Wider-scale dissemination and adoption of IPM strategies will depend on farmers' willingness to pay for the technology. Secondly, our study assesses the pre-conceived attitude, knowledge, and practices of smallholder tomato farmers regarding *T. absoluta*, which are also important pre-requisites for wider scaling of the IPM strategies for the invasive pest. Having invaded Africa recently, farmers' awareness and management strategies of the pest have not been fully understood. The findings of this study therefore will shape the policy direction on scaling up the alternative methods for management of the invasive pest.

The results show that 95% and 66% of the farmers in Kenya and Uganda were willing to adopt the proposed IPM strategy. Regarding adoption forecasting, we find that knowledge, attitude, practices, and training were the main drivers for IPM demand. The study also shows that attitude levels were generally high in Kenya than in Uganda. However, both the levels of knowledge and non-pesticide practices in Uganda were reported to be higher than those in Kenya. As expected, the application of non-pesticide practices was lower in comparison to the knowledge levels of the practices in both countries.

## 2. Materials and Methods

### 2.1. Study Area and Data Collection

The data used in this study was obtained from tomato-producing regions in Kenya and Uganda (Figures 1 and 2 respectively), the study benchmark sites. Multistage sampling was utilized. First, two counties in Kenya (Kirinyaga and Kajiado) and two districts in Uganda (Mbale and Masaka) were purposively selected due to their predominance in tomato production. These areas have a generally rich fertile soil and a favorable climate for tomato production [17,18].

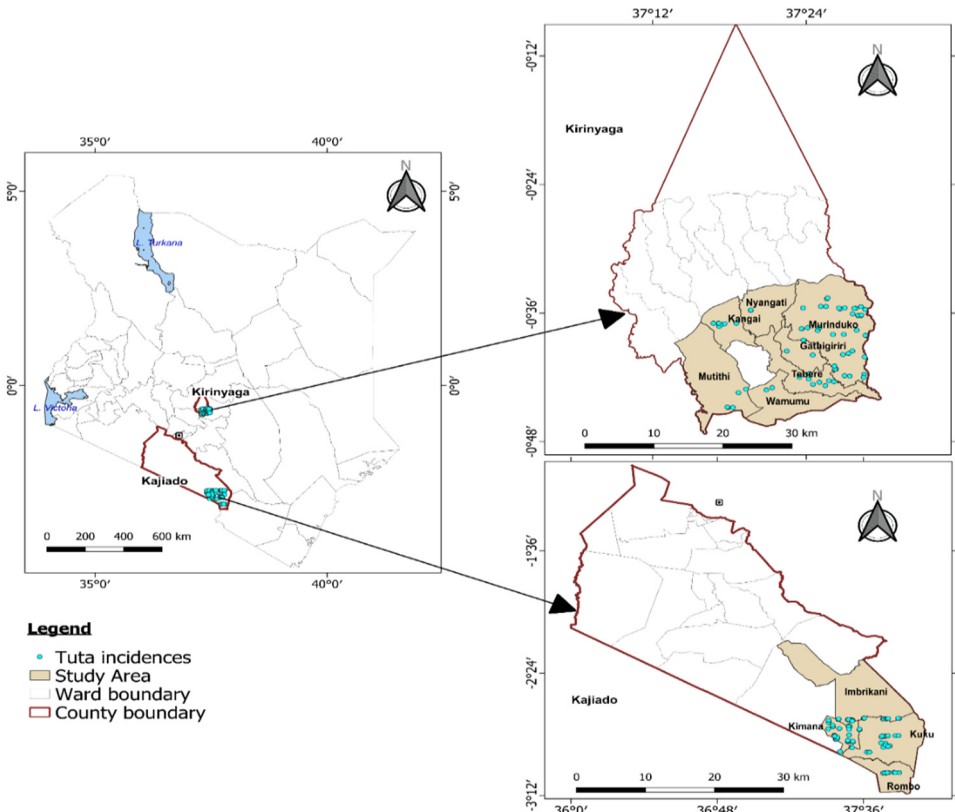

**Figure 1.** Map of Kenya showing the study sites.

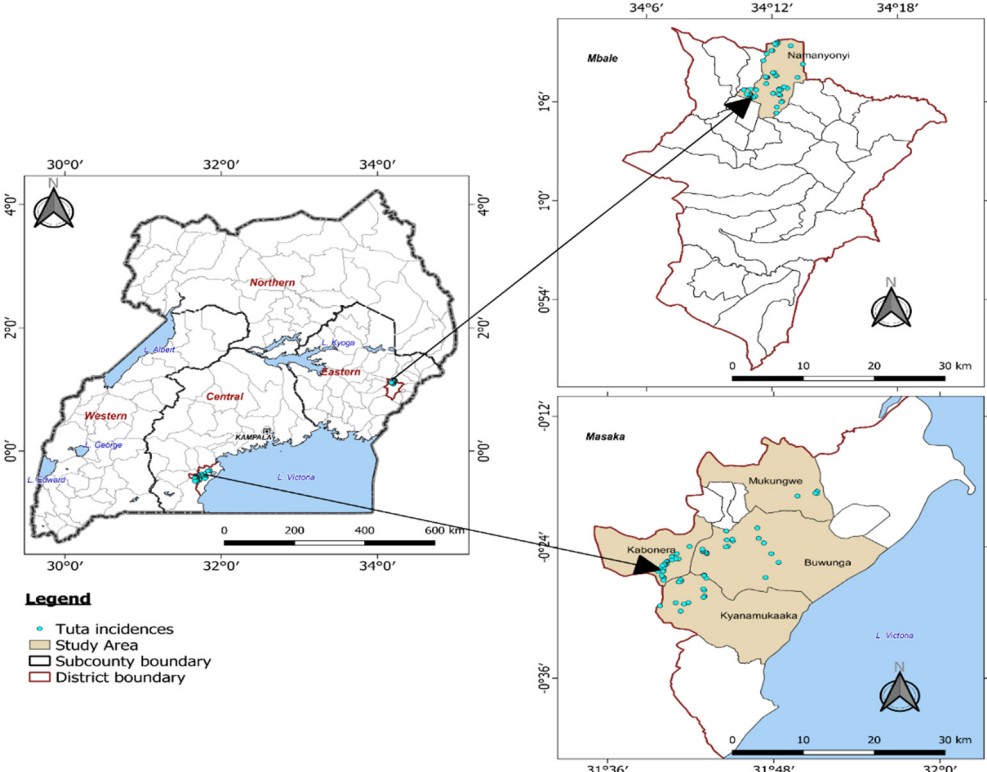

**Figure 2.** Map of Uganda showing the study areas.

In the second stage, the sub-counties and locations where tomatoes are produced were identified. A list of tomato growers from the selected sites was then developed with the

support of front-line extension workers. The lists provided sampling frames from which samples of 316 and 345 tomato growing households were randomly selected and successfully interviewed in Kenya and Uganda, respectively. The sample sizes were computed following the standard procedure [19]. A semi-structured, pre-tested questionnaire and programmed in Cspro was utilized to collect data. Data collection was carried out between June and May 2019 in both countries.

The questionnaire captured broad information on tomato production and marketing that includes farmers' knowledge and perception of tomato infesting pests and diseases, awareness, and willingness to use the proposed IPM practices, yield, marketed surplus, and other contextual characteristics. To capture the farmer's knowledge and practice, the respondents were asked about their awareness and use of various non-pesticide practices for the management of *T. absoluta* as well as symptoms associated with *T. absoluta* infestation (Table A1). A correct (yes) response was given a score of one (1) while a wrong (no) response and 'do not know' was given a score of zero (0). The respondents' knowledge and practice scores were then calculated from a total sum of correct (yes) responses. These scores were then categorized into levels, where a higher score indicated a high knowledge/good practice and a lower score indicated low knowledge/bad practice. Farmer's attitude was captured through some perception questions about *T. absoluta* (Table A1). Similarly, a correct response was coded with 1 for agreeing while the wrong response (disagree) or 'do not know' was zero (0). These scores were then summed up to give the attitude scores, which were then categorized into good/bad attitudes. The individual knowledge, attitude, and practice levels were calculated based on the above-the-mean cut-off point.

The willingness to adopt the IPM strategy was elicited based on the strategy's effects on the farmer in terms of profitability, health, and environmental effects. The farmers were subjected to a hypothetical situation consisting of three parts. First, the respondents were given the scientific background information on IPM use in comparison to synthetic chemicals including monetary and non-monetary costs and benefits associated with either. Secondly, the respondents were presented with four different production processes (using chemical alone, biopesticides alone, both approaches, and using nothing). Under each process, the respondents were informed of the cost per unit land, efficacy, health impact, and loss of biodiversity, after which, they picked the category that they would be willing to adopt. Lastly, the farmers who were willing to adopt were then asked how soon they were willing to adopt the strategy.

### 2.2. Analytical Strategy

2.2.1. Descriptive Statistics

The study employed descriptive statistics, such as percentages and frequencies. The statistical *t*-test was utilized to compare the average values of the quantitative variables between adopters and non-adopters groups, which we define in the subsequent section.

2.2.2. Potential Demand for *T. absoluta* IPM Strategy

Farmers are rational and thus seek strategies and technologies to minimize the losses and costs associated with pests. Several studies modeling the potential (*ex-ante*) adoption of agricultural technologies mainly use binary models. For instance, Groote et al. [20] used contingent valuation in estimating the potential demand of IR maize seed, while Kolady and Lesser [21] used bivariate probit to estimate the probability of the adoption of genetically modified eggplant (Bt eggplant) in India. Similar to the work of Beck and Gong [22], our study categorized the farmers into adopters and non-adopters before fitting the data into binary probit regression.

We model the adoption of IPM based on a random utility approach. Faced with two options (adopt/not adopt), a farmer is assumed to choose the alternative with the highest expected utility, and the utility of a choice depends on its attributes [23]. In our case, the non-priced inputs are the IPM technology. Let the farmer's willingness to pay for the IPM

strategy be denoted by *y*, while the alternative is denoted by *x*. Random utility categorizes utility into deterministic and random variables:

$$Ux = D_{ix} + \varepsilon_i \tag{1}$$

$$Uy = D_{iy} + \varepsilon_i \tag{2}$$

where $U$ is the utility derived from being an adopter or a non-adopter, $D_{iy}$ is a vector of explanatory variables, while $\varepsilon_i$ is the error term.

If the farmer assumes $Ux > Uy$ then the probability of the farmer choosing *x* over *y* is:

$$P_x = P\{\varepsilon_{ix} - \varepsilon_{iy} < D_{iy} - D_{ix}\}. \tag{3}$$

The probability of choosing *y* over *x* is:

$$P_y = P\{\varepsilon_{iy} - \varepsilon_{ix} < D_{ix} - D_{iy}\}. \tag{4}$$

The error terms are assumed to satisfy the property of independence from alternative variables. The probability that an individual will choose the IPM strategy is:

$$P_{iy} = \frac{e^{(D_{iy} - D_{ix})}}{1 + e^{(D_{iy} - D_{ix})}}. \tag{5}$$

The binary logit response takes two values. It is preferred since it does not assume normality, linearity, or homoscedasticity of data [24]. This model can be represented as follows linearly:

$$\text{Logit} = \ln \frac{P_i}{1 - P_i} = \beta_O + \beta_1 X_1 + \beta_2 X_1 + \beta_3 X_3 + \ldots + \beta_n X_n + \varepsilon_i \tag{6}$$

where $P_i$ is the probability of being an adopter and $1 - P_i$ is the probability of being a non-adopter. $\beta_0$ is the constant, $\beta_1 - \beta_n$ are the correlation coefficients of the independent variables while the $X_1 - X_n$ are the observable characteristics of the tomato growing household that are likely to affect adoption.

As explained in the previous section, we utilized the timing on the adoption of the IPM strategy to categorize the households into two groups. For ease of analysis, we use adopters vs. non-adopters. The non-adopters ($P_0$) refer to those who were either not planning to purchase the IPM strategy or those who were willing to purchase it later than 2 years, while the adopters ($P_1$) are the individuals planning on purchasing the IPM strategy and using it immediately or within one year. The categorization of respondents into adopters and non-adopters was based on the project timelines, which were 36 months; some of the project activities such as capacity building for farmers will not be possible after the end of the project.

The choice of the independent variables was guided by existing literature on adoption and willingness to pay for agricultural innovations, and the study context. The socioeconomic variables affecting the potential demand for IPM included household characteristics (age, sex, and education of the household head, household size, and farming experience), household resources (the proportion of income from tomato farming, farm size, livestock owned in tropical units (TLU), off-farm income, and credit constraint), and access to market and institutional information (distance to inputs, distance to the nearest trading center, distance to the nearest extension officer, and training). Besides, the knowledge, attitude, and practice with regard to tomato infesting pests were also considered as important factors that affect the potential adoption of IPM technologies. Social capital and network captured using the number of people that can be relied on, group membership, confidence in agricultural extension, and location were also considered [25–29].

To forecast *T. absoluta* adoption, Bayesian analysis was used. This method is based on the Bayes rule which assumes that the independent variables are random, therefore,

providing a number of likelihood models and prior distributions for post estimation. Selected variables, which were likely to affect demand in the short run were then fitted on a Bayesian logistic regression. These variables were chosen based on their likelihood to change following IPM strategy promotion and dissemination efforts. They include knowledge level on symptoms and non-pesticides use, practice levels on non-pesticides, attitude on the effects of *T. absoluta,* and training [29]. Sensitivity analysis was then carried out where each variable was subjected to various scenarios while the remaining variables were held constant, to get the best possible level of adoption.

### 2.2.3. Knowledge, Attitude, and Practice

A binary probit regression was used to determine the relationship between the separate KAP scores (high and low) for knowledge and (good and bad) for attitude and practice, against the socioeconomic characteristics. Let $y$ represent the individual's knowledge, attitude, and practices while $x$ represents the set of explanatory variables. The model estimates how a change in the $x$ variable causes a change in the $y$ variable. Therefore, the probit response probability can be expressed as follows:

$$prob \left( y = \frac{1}{x} \right) = \Phi \left( x\beta \right) \tag{7}$$

where $\Phi$ indicated the cumulative standard normal cumulative function, $x$ represents the independent variables, while $\beta$ is the correlation coefficients of the independent variables. The selection of the independent variables was guided by existing literature on KAP, which are similar to the ones in the adoption model above [30–37].

## 3. Results

### 3.1. Descriptive Statistics

#### 3.1.1. Farmer's Awareness and Perceived Severity of Tomato Infesting Insect Pests

To determine the awareness of tomato infesting insect pests, farmers were asked to identify the major insect pests and diseases affecting their production and the level of severity. *T. absoluta* was identified as the major pest affecting tomato production reported by about 45% and 32% of the respondents in Kenya and Uganda, respectively (Table 1). These results are supported by the findings by Nderitu et al. [38], in whose study 90% of the farmers reported *T. absoluta* as the major tomato infesting pest. Other major pests reported by the respondents included whiteflies (20.11%), red spider mite (15.11%), and thrips (14.66%) in Kenya, and cutworm (19.73%), bollworm (15.87%), and aphids (13.57) in Uganda (Table 1).

**Table 1.** Tomato infesting insect pests reported by sampled farmers in Kenya and Uganda.

| Pests | Proportion of Farmers (Percent) | |
|---|---|---|
| | Kenya | Uganda |
| Tomato Leaf Miner (*T. absoluta*) | 44.86 | 32.36 |
| Whiteflies (Aleyrodidae) | 20.11 | 9.60 |
| Red spider mite (Tetranychus urticae) | 15.11 | 2.09 |
| Thrips (Thysanoptera) | 14.66 | 6.16 |
| Cutworm (Agrotis ipsilon) | 5.23 | 19.73 |
| Bollworm (Helicoverpa armigera) | 5.11 | 15.87 |
| Aphids (Aphidoidea) | 2.95 | 13.57 |
| Other leaf miners | 0.80 | 0.00 |
| Leaf eaters | 0.34 | 0.42 |

#### 3.1.2. Farmer's Knowledge on the Tomato Leaf Miner (*T. absoluta*) Infestation Symptoms

The farmers who correctly identified *T. absoluta* were then tested on their ability to identify the symptoms of the pest damage. The two major symptoms identified were: The

pest creates mine/galleries by 23.41% (Kenya) and 44.83% (Uganda) (Table 2), and young larvae penetrate the leaves for feeding and development by 21.35% (Kenya) and 7.80% (Uganda). As observed by Shree et al. [39], these symptoms can easily be noticed from the plant leaves and fruits, which are infested at the larval stage of the pest. Approximately, 12.6% of the respondents in both countries identified female oviposit on all parts of tomatoes plant but with a preference for leaves [4]. The eggs hatch into larvae after an average of four days. The newly hatched larvae penetrate the leaves and dig mines in which they feed and develop. The older instar larvae migrate and attack other plant parts. The two least common symptoms were heavy infestation that leads to leaf defoliation and death of the plant and mining damage on the stem that causes malformation of the plant.

**Table 2.** Farmer's knowledge on symptoms of tomato *T. absoluta* infestation.

| Symptoms of *T. absoluta* | Kenya (%) | Uganda (%) |
|---|---|---|
| Pest create mines/galleries | 23.41 | 44.83 |
| Young larvae penetrate the leaves for feeding and development | 21.35 | 7.80 |
| Female oviposit on all plant parts of tomatoes with a preference for leaves | 12.64 | 12.59 |
| The pest attacks all aerial parts of the plant | 14.61 | 8.40 |
| Larvae also attack stem, young shoots, flowers, apical buds, and fruits | 16.20 | 5.70 |
| Heavy infestation leads to leaf defoliation and death of the plant | 6.84 | 10.94 |

### 3.1.3. Management Practices of the Tomato Infesting *T. absoluta*

To establish the knowledge and practice for the management of *T. absoluta*, the households were asked questions on non-pesticide practices that they were aware of and used in their tomato plots. The results are presented in Table 3. A higher percentage of tomato growers in Uganda (91%) displayed better knowledge of the non-pesticide control practices of *T. absoluta* compared to tomato farmers in Kenya (76%). In both countries, the majority of the farmers were aware of the cultural control practices, which included: Crop rotation with a non-host crop (76% Kenya and 91% Uganda); planting resistant/tolerant varieties (33% Kenya and 86% Uganda); soil tillage (2% Kenya and 83% Uganda); and picking and destroying the infected plants or plant parts (18% Kenya and 89% Uganda). This practice was reported as the most commonly known and can be attributed to its low level of technicality and its ability to reduce infestation levels [40] compared to the knowledge of biological control. Similar results were observed in a study done by Piñero & Keay [34] where cultural control practices were known by the majority of farmers while biological control was the least known. The use of parasitoids/natural enemies recorded the least awareness. The knowledge on pheromone traps for scouting, monitoring, and mass trapping and sticky traps were fairly known to both farmers in Kenya and Uganda. Farmers also all agreed that *T. absoluta* laid its eggs on all parts of the plant especially the leaves.

Non-pesticides use or practice was then measured based on the proportion of farmers who were aware of these alternatives. Crop rotation with a non-host crop was found to be the commonly used method in both countries, while the least used was biological control using parasitoids/natural enemies while the same trend applied to their level of knowledge. Cultural control methods, such as soil tillage, planting resistant/tolerant varieties, picking and destroying infected plants or plant parts, growing tomatoes under insect net or net house, selecting healthy seeds or sanitizing seed treatment, orchard sanitation (collecting fallen infested fruits and disposing away of the farm), and adjusting planting/harvesting dates and irrigation timing/amount to reduce pest damage were found to be the most commonly used type of non-pesticide practices. This could be attributed to the low costs and low level of technical skills required compared to monitoring and mass trapping (use of pheromones traps for scouting, monitoring, and mass trapping, hanging sticky traps,

and using water traps) and biological control (application of biopesticides and biological control using parasitoids/natural enemies), hence its low diffusion and use [32].

**Table 3.** Farmers' awareness and use of non-pesticide practices for the management of *T. absoluta*.

| Non-Pesticide Practice for Suppressing *T. absoluta* | Kenya | | Uganda | |
|---|---|---|---|---|
| | Aware (%) | Use (%) | Aware (%) | Use (%) |
| Planting resistant/tolerant varieties | 33 | 18 | 86 | 69 |
| Selecting healthy seeds or sanitizing seed treatment | 21 | 14 | 61 | 46 |
| Soil tillage | 25 | 25 | 83 | 75 |
| Crop rotation with non-host crop | 76 | 72 | 91 | 87 |
| Adjust planting/harvesting dates to reduce pest damage | 10 | 7 | 41 | 31 |
| Adjust irrigation timing/amount to reduce pest damage | 5 | 5 | 30 | 22 |
| Grow tomato under insect net or net house | 10 | 10 | 77 | 57 |
| Pick and destroy the infected plant or plant parts | 18 | 14 | 89 | 80 |
| Orchard sanitation | 12 | 7 | 59 | 48 |
| Use Pheromones traps for scouting, monitoring, and mass trapping | 41 | 10 | 12 | 2 |
| Hang sticky traps | 29 | 4 | 14 | 3 |
| Apply Bio pesticides | 28 | 3 | 53 | 16 |
| Biological control using parasitoids/natural enemies | 5 | 1 | 6 | 1 |
| Using a barrier crop | 4 | 2 | 18 | 8 |
| Using water traps | 9 | 4 | 9 | 4 |

3.1.4. Farmer's Response to Knowledge, Perception, and Practices towards Tomato Infesting *T. absoluta*

As shown in Table 4, the head and spouse were presented with a series of knowledge and perception statements to test their attitude towards *T. absoluta*. A higher percentage of farmers in both Kenya and Uganda had a good attitude based on the pest knowledge statements. In both countries, the head and spouse agreed that *T. absoluta* was a threat to tomato production and this affected the market value of their produce. A significant number of respondents believed that pesticides had an immediate effect on all insects (96% heads and 93% spouses) in Kenya and Uganda (75% heads and 63% spouses), while a lower percentage agreed that mixing different pesticides make them more effective (69% heads and 47% spouses) in Kenya and Uganda (53% heads and 35% spouses). The qualitative information gathered during the survey revealed pest resistance as the main driver of the use of cocktail pesticides. The findings further show limited use of government extension services as demonstrated by the small percentage of those who reported *T. absoluta* infestation to government agricultural extension officers and the effectiveness of the extension officers in offering adequate advice on the management of *T. absoluta*. However, the majority of the respondents believed that non-pesticide (IPM) practices are a better alternative to synthetic chemicals since they were concerned about the short-term and long-term health effects on animals as well as on the environment.

**Table 4.** Farmers' attitude on tomato-infesting *T. absoluta*.

| Perception/Attitude Statements (1 = Agree 0 = Otherwise) | Kenya | | Uganda | |
|---|---|---|---|---|
| | Head (%) | Spouse (%) | Head (%) | Spouse (%) |
| *T. absoluta* species are a threat to the horticulture industry | 100 | 100 | 97 | 89 |
| *T. absoluta* reduces the tomato quality | 100 | 100 | 99 | 91 |
| *T. absoluta* results in a high loss of market value | 100 | 100 | 97 | 86 |
| *T. absoluta* is a trade quarantine problem | 83 | 80 | 27 | 17 |
| *T. absoluta* eggs are laid on all plant parts of tomato with a preference for leaves | 77 | 83 | 77 | 51 |
| I prefer using pesticides that kill all insects immediately | 96 | 93 | 75 | 63 |
| I am concerned about the short-term human health effects of using pesticides e.g., headache | 88 | 93 | 78 | 73 |
| I am concerned about the long-term human health effects of using pesticides e.g., cancer | 86 | 90 | 72 | 71 |
| Synthetic chemicals present a major risk to aquatic animals, birds, mammals, and useful insects like bees. | 77 | 87 | 72 | 61 |
| Synthetic chemicals present a major risk to the surface and groundwater. | 78 | 87 | 66 | 58 |
| Mixing different pesticides can make them more effective | 69 | 47 | 53 | 38 |
| The spread of *T. absoluta* can be prevented | 61 | 73 | 88 | 72 |
| Non-pesticide (IPM) are better alternatives to synthetic chemicals | 60 | 67 | 61 | 49 |
| Chemical pesticides alone can effectively control *T. absoluta* | 48 | 43 | 55 | 33 |
| Adult *T. absoluta* do not feed on fruits | 33 | 37 | 14 | 44 |
| Report *T. absoluta* infestation to gov. agric. extension officers | 32 | 30 | 37 | 43 |
| Extension officers offer adequate advice on the management of *T. absoluta* | 24 | 33 | 27 | 25 |

### 3.1.5. Potential Adoption of *Tuta absoluta* IPM Strategies

Table 5 shows the potential *T. absoluta* IPM adoption patterns among tomato farmers in Kenya and Uganda based on the willingness to pay responses according to the year they were willing to start using the IPM strategy. Five (5%) and 33% of the respondents are classified as non-adopters, while 95% and 66% are classified as adopters in Kenya and Uganda, respectively.

**Table 5.** Potential adoption patterns of the tomato leaf miner IPM technologies/strategies by tomato farmers in Kenya and Uganda.

| | Kenya | | Uganda | |
|---|---|---|---|---|
| | N | % | N | % |
| Non adopters | 16 | 5 | 116 | 33 |
| Adopters | 300 | 95 | 229 | 66 |
| Total | 316 | 100 | 345 | 100 |

### 3.1.6. Selected Socio-Economic Characteristics That Influence KAP and IPM Adoption

Selected socio-economic characteristics of the sampled households are presented in Table 6. A *t*-test was carried out to determine the mean difference of the explanatory variables between the adopters and non-adopters. Regarding the household characteristics, the gender of the household head was significantly different, at the 5 percent level ($p < 0.05$) between IPM adopters and non-adopters in the two study countries. The two categories

showed a higher percentage willing to adopt the IPM strategy, which differs from existing literature that shows that there is no significant difference between the rate of adoption and the gender of the household head [26].

With respect to household resources, livestock owned in tropical livestock units and access to off-farm income was found to be significantly different ($p < 0.05$) between IPM adopters and non-adopters in Kenya and Uganda, respectively. This difference in off-farm income can be attributed to the fact that households with access to alternative income have more disposable income for adoption compared to those without [25,28].

With reference to access to market and institutional information, distance to the nearest source of inputs and training was significantly different ($p < 0.001$) between IPM adopters and non-adopters in both countries. The expected effect of the distance is however indeterminate since IPM adopters in Kenya reported longer distances while those in Uganda reported shorter distances. Training was found to be significantly different ($p < 0.05$) between IPM adopters and non-adopters. In Kenya distance to the nearest government extension officer was also found to be significantly different ($p < 0.001$) between adopters and non-adopters. The difference in the market and institutional information can be attributed to access to information. Farmers with easier access to information are more likely to be adopters compared to those with limited access [41].

With respect to knowledge, attitude, and practice levels, the attitude level in both countries was found to be significantly different between adopters and non-adopters. The finding corroborates with existing literature [42–44].

### 3.2. Empirical Results

#### 3.2.1. Factors Affecting Potential Demand for *T. absoluta* IPM

Explanatory variables described in Table 6 above were used to model the potential demand for *T. absoluta* IPM. The model results are presented in Table 7. In Kenya, the potential demand for the IPM was significantly related to distance to inputs, training, knowledge, and practice levels, while in Uganda the potential demand was correlated with gender, distance to the nearest government extension agricultural office, and attitude score. Gender had a positive relationship with the potential demand in Uganda, implying that households headed by males are more likely to be early adopters compared to female-headed households. This may be because women farmers tend to have access to fewer resources compared to male farmers. Distance to inputs suppliers was found to have a positive relationship with adoption, suggesting that households located further from the source of inputs are less likely to be adopters compared to those nearby [29]. Long distances to input suppliers increase transaction costs through the transportation of inputs. The training was also found to be positively related to IPM adoption. Farmers who attended the training were more likely to be early adopters compared to farmers who never attended the training.

**Table 6.** Selected farm and farmer characteristics of tomato farmers in Kenya and Uganda.

| | Kenya | | | | | Uganda | | | | |
| --- | --- | --- | --- | --- | --- | --- | --- | --- | --- | --- |
| | Non Adopters | | Adopters | | | Non-Adopters | | Adopters | | |
| | Mean | Standard Error | Mean | Standard Error | *t*-Tests | Mean | Standard Error | Mean | Standard Error | *t*-Tests |
| Household characteristics | | | | | | | | | | |
| Age (years) | 43.438 | 2.875 | 43.840 | 0.663 | 0.892 | 42.914 | 1.140 | 42.621 | 0.793 | 0.833 |
| Gender (1 = Male, 0 = Female) | 0.875 | 0.120 | 0.927 | 0.194 | 0.040 ** | 0.750 | 0.0403 | 0.860 | 0.023 | 0.011 ** |
| Education (years) | 9.313 | 0.723 | 8.680 | 0.201 | 0.476 | 7.226 | 0.358 | 7.181 | 0.223 | 0.911 |
| Household size (adult equivalent) | 2.244 | 0.233 | 2.346 | 0.441 | 0.607 | 2.772 | 0.085 | 2.905 | 0.067 | 0.238 |
| Farming Experience (years) | 12.250 | 2.116 | 12.633 | 0.466 | 0.854 | 10.543 | 0.801 | 10.934 | 0.638 | 0.713 |
| Household resources | | | | | | | | | | |
| Proportion of income from tomatoes (%) | 44.563 | 6.299 | 48.393 | 1.414 | 0.543 | 39.509 | 1.735 | 37.544 | 1.405 | 0.402 |
| Total Farm size (acres) | 2.789 | 0.748 | 4.096 | 0.326 | 0.360 | 3.445 | 0.483 | 3.349 | 0.372 | 0.878 |
| Livestock owned in Tropical Livestock Units (TLU) | 1.818 | 0.448 | 3.617 | 0.186 | 0.569 ** | 1.062 | 0.150 | 1.174 | 0.117 | 0.510 |
| Credit constraint (dummy) | 0.250 | 0.125 | 0.363 | 0.028 | 0.947 | 0.466 | 0.047 | 0.491 | 0.033 | 0.625 |
| Have access to off-farm income (dummy) | 0.125 | 0.085 | 0.280 | 0.260 | −0.176 | 0.707 | 0.042 | 0.576 | 0.032 | 0.018 ** |
| Access to market and institutional information | | | | | | | | | | |
| Distance to the nearest inputs center [a] | 23.750 | 4.503 | 45.480 | 2.569 | 0.053 | 115.069 | 6.9226 | 68.825 | 4.251 | 0.000 *** |
| Distance to the nearest trading center [a] | 22.750 | 4.085 | 37.977 | 2.299 | 0.129 | 18.089 | 1.558 | 24.625 | 1.446 | 0.245 |
| Distance to the nearest agricultural office [a] | 172.938 | 34.903 | 92.647 | 5.070 | 0.000 *** | 87.655 | 8.778 | 84.947 | 4.396 | 0.758 |
| Attended training (dummy) | 0.188 | 0.101 | 0.457 | 0.288 | 0.035 ** | 0.220 | 0.038 | 0.392 | 0.032 | 0.001 ** |
| Knowledge, attitude, and practices | | | | | | | | | | |
| Knowledge level (score) | 0.250 | 0.112 | 0.407 | 0.028 | 0.214 | 0.371 | 0.045 | 0.572 | 0.032 | 0.000 *** |
| Attitude level (score) | 0.250 | 0.112 | 0.687 | 0.027 | 0.000 *** | 0.578 | 0.046 | 0.677 | 0.031 | 0.069 ** |
| Practice level (score) | 0.250 | 0.112 | 0.453 | 0.029 | 0.111 | 0.603 | 0.046 | 0.511 | 0.501 | 0.104 |
| Social capital and networks | | | | | | | | | | |
| Tomato group membership (dummy) | 0.000 | 0.008 | 0.020 | 0.008 | 0.569 | 0.009 | 0.008 | 0.022 | 0.010 | 0.374 |
| Number of people that can be relied on in critical needs (number) | 5.188 | 1.385 | 5.370 | 0.461 | 0.928 | 3.784 | 0.247 | 3.930 | 0.178 | 0.634 |
| Confidence in extension officers (dummy) | 0.188 | 0.101 | 0.270 | 0.026 | 0.468 | 0.310 | 0.043 | 0.246 | 0.029 | 0.200 |
| Location Dummies | | | | | | | | | | |
| Kirinyaga and Kajiado County | 0.939 | 0.128 | 0.959 | 0.288 | 0.425 | | | | | |
| Mbale and Masaka districts | | | | | | 1.362 | 0.044 | 1.539 | 0.033 | 0.001 ** |

Note: Source: Household survey; ** *p* < 0.05; *** *p* < 0.01; [a] All distances in walking minutes; dummy represent 1 = yes, 0 = otherwise.

**Table 7.** Factors influencing potential demand of IPM strategy among smallholder farmers in Kenya and Uganda.

| | Kenya | | | Uganda | | |
|---|---|---|---|---|---|---|
| | Coefficients | Standard Error | Marginal Effects | Coefficients | Standard Error | Marginal Effects |
| **Dependent Variable Potential Demand** | | | | | | |
| *Household characteristics* | | | | | | |
| Age | 0.000 | 0.021 | 0.000 | 0.001 | 0.008 | 0.000 |
| Gender | 1.058 | 0.447 | 0.022 | 0.246 * | 0.209 | 0.079 |
| Education | −0.100 | 0.062 | −0.001 | −0.034 | 0.025 | −0.012 |
| Household size (adult equivalent) | 0.005 | 0.282 | 0.000 | 0.085 | 0.090 | 0.028 |
| Experience | 0.001 | 0.027 | 0.000 | −0.002 | 0.009 | 0.000 |
| *Household resources* | | | | | | |
| Proportion of income from tomatoes | 0.002 | 0.008 | 0.000 | −0.002 | 0.004 | −0.002 |
| Total Farm size | 0.105 | 0.079 | 0.001 | 0.007 | 0.023 | 0.000 |
| Livestock owned in Tropical Livestock Units (TLU) | −0.003 | 0.012 | 0.000 | −0.009 | 0.053 | −0.002 |
| Credit constraint | −0.098 | 0.374 | −0.001 | −0.030 | 0.169 | 0.000 |
| Have access to off-farm income | 0.677 | 0.53 | 0.004 | −0.246 | 0.174 | −0.090 |
| *Access to market and institutional information* | | | | | | |
| Distance to inputs | 0.014 ** | 0.014 | 0.000 | −0.004 | 0.001 | −0.002 |
| Distance to the nearest agricultural extension office | −0.004 | 0.002 | 0.000 | −0.001 ** | 0.001 | 0.000 |
| Attended training | 0.329 ** | 0.387 | 0.002 | 0.498 | 0.183 | 0.158 |
| *Knowledge attitude and practices* | | | | | | |
| Knowledge level | −0.316 ** | 0.612 | −0.003 | 0.626 | 0.193 | 0.190 |
| Attitude level | 1.158 | 0.421 | 0.014 | 0.144 ** | 0.176 | 0.038 |
| Practice level | 1.059 ** | 0.631 | 0.007 | −0.509 | 0.187 | −0.153 |
| *Social capital and networks* | | | | | | |
| Number of people that can be relied on in critical needs | −0.01 | 0.018 | 0.000 | 0.008 | 0.007 | 0.003 |
| Confidence in extension officers | 0.165 | 0.42 | 0.001 | −0.225 | 0.190 | −0.192 |
| *Location Dummies* | | | | | | |
| Kirinyaga and Kajiado County | 0.153 | 0.411 | | | | |
| Mbale and Masaka districts | | | | 0.243 | 0.208 | 0.343 |
| Constant | −0.427 | 1.49 | | 0.249 | 0.665 | |
| Number of observations | 316 | | | 343 | | |
| LR chi2(40) | 44.52 | | | 71.310 | | |
| Pseudo R2 | 0.3516 | | | 0.163 | | |
| Log pseudo likelihood | −41.055981 | | | −183.801 | | |

Note: Source: Household survey; * $p < 0.1$; ** $p < 0.05$.

In Uganda, distance to the nearest government agricultural extension office was found to be negatively related to the potential demand for the IPM strategy. These findings were contrary to our expected hypothesis as it suggests that farmers located further from the agricultural extension offices were more likely to be early adopters compared to those located close by [27].

Contrary to our expectation, knowledge level was found to have a negative relation with potential demand for IPM. This could be explained by the risk averseness of farmers regarding trying a new product such as IPM, as they are well conversant with the damage caused by the *T. absoluta*. The practice level was positively correlated with the potential demand in Kenya, implying that farmers using good practices (i.e., currently using some elements of the IPM) were more likely to be early adopters compared to those who were using conventional practices (synthetic chemicals). The attitude score had a positive relationship with the potential IPM adoption in Uganda, suggesting that farmers having

a good attitude were more likely to be early IPM adopters compared to those without a similar attitude. The attitudes scores were based on the symptoms, occurrence, and effects of *T. absoluta* on tomato production.

### 3.2.2. Factors Affecting KAP towards the Tomato Leaf Miner (*T. absoluta*) Management

As presented earlier in the descriptive summary (Table 6), with respect to KAP, 40% of the respondents from Kenya and 50% from Uganda were above the mean on the knowledge score, indicating good knowledge. Similarly, 67% and 64% of the respondents from Kenya and Uganda, respectively had an attitude score above the mean, and therefore a good perception, while 44% and 54% of the respondents in Kenya and Uganda, respectively had a non-pesticide practice score above the mean.

Table 8 presents the binary regression results that show the relationship between the individual KAP variables and the selected socioeconomic characteristics of the respondents. With respect to knowledge, results show that in Kenya, the male-headed households had a positive correlation to good knowledge level compared to female-headed households. The finding is affirmed by a study done by Atreya [30], which showed that pest management decisions are male-dominated. In both countries, education was found to be significant. In Uganda, the level of education was positively related to the level of knowledge compared to Kenya where it was found to be negative. Individuals with higher levels of education were found to have lower knowledge of *T. absoluta*. Harapan et al. [35] support the results from Uganda where they found the odds of having good knowledge was correlated with the level of education. In Kenya, distance to the nearest trading center and the government agricultural office were found to be negatively correlated with the knowledge score. In Uganda, distance to inputs was negatively correlated to the knowledge level while training was positively related to the knowledge score. Training leads to improved knowledge [36]. Location dummies in both countries were found to be negatively related to the knowledge score.

In Kenya, gender, training, and knowledge score were found to be positively related to the attitude level. Distance to the nearest government office was found to be negatively related to the attitude level. In Uganda, age was found to be negatively related while the farming experience was positively related to attitude level. Location dummy was found to be negatively related to the attitude level.

Regarding the practice level, in Kenya, knowledge level was found to be positively related to the practice scores as individuals will practice what they have more knowledge on [37]. In Uganda distance to the nearest trading center and training was found to be negatively related to the practice level.

### 3.2.3. Marginal Effects

The marginal effect shows the impact on the covariate as a result of a change in the independent variables. The marginal effects were calculated as shown in Table 7. In reference to household characteristics, male-headed households increased the probability of IPM adoption by 7.9% in Uganda. One percent increase in training in Kenya had a 0.2% effect on the probability of IPM adoption. In relation to knowledge, attitude, and practices, a 1% increase in knowledge results in a 0.3% increase in the probability of IPM non-adoption in Kenya. In respect to perception/attitude, a percent increase leads to a 3.8% increase in IPM adoption probability in Uganda whereas a 1% increase in practices causes a 0.7% increase in IPM adoption in Kenya.

**Table 8.** Binary probit model showing the relationship between knowledge attitude and practices on socioeconomic characteristics.

| | Knowledge Level | | | | Attitude Level | | | | Practice Level | | | |
|---|---|---|---|---|---|---|---|---|---|---|---|---|
| | Kenya | | Uganda | | Kenya | | Uganda | | Kenya | | Uganda | |
| | Coef. | SE | Coef. | SE | Coef. | SE | Coef. | SE | Coef. | SE | Coef. | SE |
| Household characteristics | | | | | | | | | | | | |
| Age | 0.010 | 0.013 | 0.003 | 0.011 | −0.001 | 0.0140 | −0.006 | 0.012 | 0.012 | 0.021 | −0.011 | 0.012 |
| Gender | 0.355 | 0.376 | 0.617 | 0.325 | 1.614 *** | 0.379 | 0.504 | 0.325 | 0.948 | 0.651 | −0.156 | 0.338 |
| Education | −0.024 * | 0.040 | 0.108 ** | 0.037 | 0.043 | 0.042 | 0.105 ** | 0.039 | 0.043 | 0.061 | 0.013 | 0.040 |
| Household size (adult equivalent) | −0.036 | 0.185 | 0.168 | 0.136 | −0.033 | 0.190 | 0.182 | 0.143 | −0.072 | 0.284 | −0.111 | 0.147 |
| Experience | −0.014 | 0.018 | 0.018 | 0.014 | 0.024 | 0.020 | 0.034 * | 0.015 | 0.024 | 0.028 | −0.004 | 0.015 |
| Household resources | | | | | | | | | | | | |
| Proportion of income from tomatoes | −0.008 | 0.005 | 0.001 | 0.006 | −0.009 | 0.006 | −0.003 | 0.006 | 0.008 | 0.009 | 0.006 | 0.007 |
| Total farm size | −0.023 | 0.026 | 0.027 | 0.035 | −0.007 | 0.027 | −0.010 | 0.027 | 0.004 | 0.040 | 0.024 | 0.031 |
| Livestock owned in Tropical Livestock Units (TLU) | −0.002 | 0.004 | −0.100 | 0.076 | 0.001 | 0.005 | 0.024 | 0.079 | −0.003 | 0.007 | 0.061 | 0.092 |
| Credit constraint | | | | | | | | | −0.135 | 0.428 | 0.172 | 0.270 |
| Access to market and institutional information | | | | | | | | | | | | |
| Distance to inputs | 0.005 | 0.004 | −0.004 * | 0.002 | 0.006 | 0.004 | 0.002 | 0.002 | 0.004 | 0.005 | 0.003 | 0.002 |
| Distance to the nearest trading Centre | −0.020 *** | 0.006 | 0.003 | 0.004 | −0.006 | 0.004 | −0.005 | 0.004 | −0.005 | 0.007 | −0.011 * | 0.005 |
| Distance to the nearest agricultural office | −0.004 * | 0.002 | 0.001 | 0.002 | −0.003 * | 0.002 | −0.001 | 0.002 | 0.000 | 0.002 | 0.003 | 0.002 |
| Attended training | −0.381 | 0.264 | 0.565 * | 0.261 | 0.785 ** | 0.286 | 0.198 | 0.281 | −0.247 | 0.412 | −0.585 * | 0.286 |
| Knowledge attitude and practices | | | | | | | | | | | | |
| Knowledge score | | | | | 0.585 * | 0.289 | 0.488 | 0.264 | 4.788 *** | 0.488 | 2.069 *** | 0.290 |
| Attitude score | | | | | | | | | 0.236 | 0.437 | −0.339 | 0.290 |
| Practice score | | | | | | | | | | | | |
| Social capital and networks | | | | | | | | | | | | |
| Tomato group membership (1 = Yes, 0 = No) | 1.005 | 0.945 | 0.828 | 0.977 | −1.222 | 0.959 | 1.456 | 1.223 | −0.388 | 1.340 | 0.931 | 1.145 |
| Confidence in extension officers | −0.345 | 0.299 | 0.161 | 0.282 | 0.208 | 0.312 | 0.067 | 0.295 | 0.196 | 0.456 | 0.349 | 0.301 |
| Location Dummies | | | | | | | | | | | | |
| Kirinyaga and Kajiado County | −0.717 * | 0.285 | | | −0.226 | 0.311 | | | 0.040 | 0.453 | | |
| Mbale and Masaka districts | | | 0.570 | 0.291 | | | 1.052 *** | 0.315 | | | 0.066 | 0.343 |
| Constant | 1.759 | 0.954 | −2.956 ** | 0.990 | −0.903 | 1.020 | −2.839 ** | 1.011 | −4.358 ** | 1.612 | −0.416 | 1.022 |
| Number of observations | 316 | | 343 | | 316 | | 343 | | 316 | | 343 | |
| LR chi2(40) | 43.770 | | 40.880 | | 51.470 | | 45.670 | | 236.260 | | 76.730 | |
| Pseudo R2 | 0.103 | | 0.091 | | 0.128 | | 0.108 | | 0.544 | | 0.173 | |
| Log pseudolikelihood | −190.623 | | −205.370 | | −175.861 | | −188.783 | | −98.849 | | −183.396 | |

Note: Source: Household survey; * $p < 0.1$; ** $p < 0.05$; *** $p < 0.01$.

### 3.2.4. Forecasting Adoption

The means from the Bayesian analysis are slightly higher compared to the baseline means as shown in Table 9 and based on a default prior normal (0,10000). Using a 95% confidence interval, the mean probability of a farmer's knowledge, attitude, practice, and training likely to influence demand is 0.322, 1.734, 1.164, and 1.267, respectively in Kenya. While in Uganda, the mean probability is 1.118, 0.907, 0.249, and 0.703 for knowledge, attitude, practice, and training, respectively. Showing that the selected variables have a higher probability of improving the adoption decision of the farmer, from being a non-adopter to an adopter.

**Table 9.** Bayesian logit analysis and conditional probability on adoption by key sensitive variables.

| | Baseline Means | | Bayesian Analysis Means | | Conditional Probability Margins | |
|---|---|---|---|---|---|---|
| | **Kenya** | **Uganda** | **Kenya** | **Uganda** | **Kenya** | **Uganda** |
| Knowledge level | 0.399 | 0.504 | 0.322 | 1.118 | 0.963 | 0.786 |
| Attitude level | 0.665 | 0.643 | 1.734 | 0.907 | 0.976 | 0.690 |
| Practice level | 0.443 | 0.542 | 1.164 | 0.249 | 0.979 | 0.639 |
| Training | 0.443 | 0.322 | 1.267 | 0.703 | 0.984 | 0.774 |

Forecasting was done by comparing the means of the existing scenario to that of a hypothetical one. We hypothesized increase in the knowledge level of the farmers to 100%, attitude level to 80%, practice level to 90%, and training to 100% as a result of the ongoing tomato IPM promotion and dissemination efforts.

As shown in Table 9, in the short run, knowledge increases the probability of IPM demand by 96% in Kenya, and by 79% in Uganda. Similarly, increase in the perception level, practice level, and training would lead to a significant increase in adoption probability. As seen previously in Table 8, the level of these selected variables in reference to farmers in the two countries has an impact on the potential adoption. Knowledge on non-pesticide practices has a great impact on their use, as also observed by Clausen et al. [45] in their study that showed reduced use of pesticides due to increased diffusion of knowledge on IPM in Uganda. This is further supported by Gautam et al. [46], who showed that training vegetable farmers in Bangladesh led to the proper use of synthetic pesticides and improved IPM adoption resulting in higher yields and an improved gross margin.

## 4. Conclusions and Policy Implications

This study aimed at assessing the knowledge, attitude, and perceptions of tomato growers in regard to invasive tomato-infesting leaf miner (*T. absoluta*), and estimate the potential adoption of a proposed IPM strategy for the suppression of the pest, using a case of Kenya and Uganda. We classified farmers as non-adopters and adopters based on the willingness to adopt and the agility of using it. Results from both descriptive and empirical analysis show there is a significant difference in the two groups (adopters and non-adopters).

Results indicate that the major tomato infesting pest in both countries is *T. absoluta*. The main management practice of this pest is the use of synthetic pesticides with the majority of respondents in both countries not aware of the negative effects of these chemical pesticides and a smaller percentage not aware of the *T. absoluta* IPM strategy. The major factors affecting the farmer's potential adoption for the *T. absoluta* IPM strategy are the gender of the household head, distance to inputs source and the nearest agricultural extension office, access to training, and knowledge, attitude, and practice levels. The study showed at least a 65% probability of adopting the IPM strategy in both countries.

Our results on adoption forecasting provide key insights into the existing literature on the adoption of agricultural innovations, such as IPM. To improve adoption in the short run, it is important to direct our attention to improving the KAP as an important driver to farmers' adoption decisions. This can be done by creating awareness, increasing the dissemination programs and training. Subsequently, this would result to increased knowledge on the infestation of *T. absoluta* and management of the pest using the sustainable IPM strategy and improved attitude on prevention and management of *T. absoluta*.

It is important to note that even though our results provide statistical evidence on the potential adoption gaps and factors influencing adoption in both Kenya and Uganda, these results should be interpreted with caution since the data is cross-sectional in nature, hence it might not be able to give a long-term picture. We, therefore, recommend further studies to be done using panel data across different regions where promotion and dissemination of the IPM technology are being carried out.

**Author Contributions:** B.W.M., S.A.M., S.N. and F.M.K. conceptualized the study and wrote the project proposal. B.W.M., F.C. and J.L. participated in the methodology and the formal data analysis. All authors have read and agreed to the published version of the manuscript.

**Funding:** This research was funded by the African Union (AU) (grant no.: AURG II-2-123-2018), and Biovision Foundation Project (grant no.: BV DPP-012/2019-2022). We also acknowledge the International Centre of Insect Physiology and Ecology (icipe) core funding from the UK's Foreign, Commonwealth, and Development Office (FCDO); the Swedish International Development Cooperation Agency (Sida); the Swiss Agency for Development and Cooperation (SDC); the Federal Democratic Republic of Ethiopia; and the Government of the Republic of Kenya.

**Institutional Review Board Statement:** Ethical review and approval were waived for this study, due to the long-standing working of the International Centre for Insect Physiology and Ecology in the study sites on various insect crops. However, oral consent was sort from the respondents who were provided with sufficient information about the research to allow them to make informed and free decisions on their participation in the study.

**Informed Consent Statement:** Informed consent was obtained from all subjects involved in the study.

**Data Availability Statement:** The data presented in this study are available on request from the corresponding author.

**Acknowledgments:** We wish to express our gratitude towards tomato growers from Kenya and Uganda who volunteered their time to participate in the survey and the enumerators for their effort in data collection. An earlier version of this paper was presented at the 31st International Conference of Agricultural Economists, 17–31 August 2021, Online. We are grateful for the helpful comments from two anonymous referees and participants of this conference.

**Conflicts of Interest:** The authors voiced that they have no conflict of interest.

## Appendix A

**Table A1.** Knowledge, attitude, and practice statements.

| Knowledge | Statements |
|---|---|
| | Symptoms of Tomato leaf miner (*Tuta absoluta*) |
| | 1. Create mines/galleries |
| | 2. Young larvae penetrate the leaves for feeding and development |
| | 3. Female oviposits on all plant parts of tomatoes with a preference for leaves |
| | 4. The pest attacks all aerial parts of the plant |
| | 5. Larvae also attack stem, young shoots, flowers, apical buds, and fruits |
| | 6. Heavy infestation leads to leaf defoliation and death of the plant |
| | 7. Mining damage on the stem causes malformation of the plant |

**Table A1.** *Cont.*

| Knowledge | Statements |
|---|---|
| | Non-pesticide management practices |
| | 1. Crop rotation with a non-host crop |
| | 2. Planting resistant/tolerant varieties |
| | 3. Soil tillage |
| | 4. Pick and destroy the infected plant or plant parts |
| | 5. Apply biopesticides (e.g., neem et al.) |
| | 6. Selecting healthy seeds or sanitizing seed treatment |
| | 7. Grow tomato under insect net or net house |
| | 8. Orchard sanitation (collecting fallen infested fruits and disposing away of the farm) |
| | 9. Adjust planting/harvesting dates to reduce pest damage |
| | 10. Use pheromones traps for scouting, monitoring, and mass trapping |
| | 11. Hang sticky traps |
| | 12. Adjust irrigation timing/amount to reduce pest damage |
| | 13. Using a barrier crop |
| | 14. Using water traps |
| | 15. Other non-pesticide control methods (specify) |
| | 16. Biological control using parasitoids/natural enemies |
| Practices (for non-pesticide *T. absoluta* management practices) | Non-pesticide management practices |
| | 1. Crop rotation with a non-host crop |
| | 2. Planting resistant/tolerant varieties |
| | 3. Soil tillage |
| | 4. Pick and destroy the infected plant or plant parts |
| | 5. Apply biopesticides (e.g., neem et al.) |
| | 6. Selecting healthy seeds or sanitizing seed treatment |
| | 7. Grow tomato under insect net or net house |
| | 8. Orchard sanitation (collecting fallen infested fruits and disposing away of the farm) |
| | 9. Adjust planting/harvesting dates to reduce pest damage |
| | 10. Use pheromones traps for scouting, monitoring, and mass trapping |
| | 11. Hang sticky traps |
| | 12. Adjust irrigation timing/amount to reduce pest damage |
| | 13. Using a barrier crop |
| | 14. Using water traps |
| | 15. Other non-pesticide control methods (specify) |
| | 16. Biological control using parasitoids/natural enemies |

**Table A1.** *Cont.*

| Knowledge | Statements |
|---|---|
| Perception | Perception Statements |

1. *T. absoluta* species are a threat to the horticulture industry
2. *T. absoluta* reduces the tomato quality
3. *T. absoluta* results to a high loss of market value
4. *T. absoluta* is a trade quarantine problem
5. *T. absoluta* eggs are laid on all plant parts of tomato with a preference for leaves
6. I prefer using pesticides that kill all insects immediately
7. I am concerned about the short-term human health effects of using pesticides e.g., headache
8. I am concerned about the long-term human health effects of using pesticides such as cancer
9. Synthetic chemicals present a major risk to aquatic animals, birds, mammals, and useful insects like bees.
10. Synthetic chemicals present a major risk to the surface and groundwater.
11. Mixing different pesticides can make them more effective
12. Spread of *T. absoluta* can be prevented
13. Non-pesticide (IPM) is a better alternative to synthetic chemicals
14. Chemical pesticides alone can effectively control *T. absoluta*
15. Adult *T. absoluta* do not feed on fruits
16. Report *T. absoluta* infestation to gov. agric. extension officers
17. Extension officers offer adequate advice on the management of *T. absoluta*

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
