# Peer review of "Knowledge, Attitude, and Practices on Tomato Leaf Miner, Tuta absoluta on Tomato and Potential Demand for Integrated Pest Management among Smallholder Farmers in Kenya and Uganda"

_agriculture, doi:10.3390/agriculture11121242_

Round 1

Reviewer 1 Report

I consider that it is a very important work in terms of obtaining the necessary information to highlight the reality that the farmer lives in relation to his crops and how to manage these crops.

It is the first study to estimate potential demand for the proposed IPM approaches for the management of this important pest in these two countries.

The study is well designed, with the methods perfectly used based on the information tha you want to obtain. Similarly, the analysis of the data obtained is correct.

Author Response

Thank you for taking the time to read our paper and for the useful feedback that has improved our paper. Below we list your comment and our responses. We hope you will find our responses acceptable.

Response to Reviewer 1 Comments

 Point 1: General: English language and style are fine/minor spell check required

Response 1: Thank you for the positive response and the suggestions. We have thoroughly read through the paper for spell check.

Comments and Suggestions for Authors:

Point 2: I consider that it is a very important work in terms of obtaining the necessary information to highlight the reality that the farmer lives in relation to his crops and how to manage these crops. It is the first study to estimate potential demand for the proposed IPM approaches for the management of this important pest in these two countries. The study is well designed, with the methods perfectly used based on the information that you want to obtain. Similarly, the analysis of the data obtained is correct.

Response 2: Thank you for the positive response. We are glad you found our study well designed and data analysis correctly conducted and the results relevant to the existing literature.

Point 3: Specific comments appended in the manuscript

Response 3: Thank you for the additional suggestions and comments that you appended in the main document. We have carefully reviewed and addressed them. 

Point 4 : Line 95-102 (results) should not be in the introduction

Response 4: Thank you for this observation. We reviewed a few papers published in this journal that follow the same introduction order and are further motivated by Evans, 2020. As a result, we would like to keep the results in the introduction so that our readers know what to expect later in the main result section based on our study objectives.

Reviewer 2 Report

This study aimed at assessing the knowledge, attitude, and perceptions of tomato  growers in regard to invasive tomato-infesting leaf miner (T. absoluta), in  Kenya and Uganda.

The composition of mansucript is clear and easy for understanding. But should be improve the table form and equation form.

 Does the local farmer concerned the pesticide resistance ? Please add some information in discussion section.

Author Response

Response to Reviewer 2 Comments  for paper

“Knowledge, Attitude, and Practices on Tomato Leaf 3 Miner, Tuta absoluta on Tomato and Potential Demand 4 for Integrated Pest Management among Smallholder 5 Farmers in Kenya and Uganda”

 Thank you for taking the time to read our paper and for the useful feedback that has improved our paper. Below we list your comment and our responses. We hope you will find our responses adequate.

Point 1: General: English language and style are fine/minor spell check required

Response 1: Thank you for the positive response and the suggestions. We have thoroughly read through the paper for spell check.

Comments and Suggestions for Authors

Point 2: This study aimed at assessing the knowledge, attitude, and perceptions of tomato growers in regard to invasive tomato-infesting leaf miner (T. absoluta), in Kenya and Uganda. The composition of manuscript is clear and easy for understanding. But should be improve the table form and equation form.

Response 2: Thank you for your comment. We have improved the table and equation presentations. We hope you will find their appearance better.

Point 3: Does the local farmer concerned the pesticide resistance? Please add some information in the discussion section.

Response 3: Thank you for your positive. Yes, the farmers are indeed concerned. This is reflected in Table 4 (Farmers' attitudes on Tomato infesting T. absoluta), where a significant number of respondents believed that mixing different pesticides makes them more effective. Qualitative information that followed this question revealed that this was practiced to manage the pest due to resistance to certain pesticides. We have added a sentence in this section to emphasize this point (see Page 9, 263-264).

Point 4: Specific comments appended in the manuscript

Response 4: We appreciate the additional corrections, suggestions, and comments you appended in the manuscript. However, we would wish to highlight a few of them below

  1. Comment [1] & Commented [M3]: add the standard error on the data

Response M3: The answers to these questions are yes/no (dummy variables, i.e. 1=Yes, 0=No). Rather than means, the figures given are the proportion of “Yes” answers. Subsequently, we do not calculate the SE.

  1. Commented [M4]: please explain this approach in the method on t-test

Response: This is a standard statistical test that is used to compare the means of two groups. We have also added a sections on descriptive analysis that highlights the use of t-test for descriptive statistics (see section 2.2.1).

  • Commented [M5]: add T and P values on the T-test in here

Response M5: We use the stars (*) to represent P-values as explained by the note below Table 6.  We do not include the p-values to avoid crowding the table and as the standard way used in most manuscripts.

  1. Commented [M7]: THIS IS P value?

Response: Yes, as explained in (ii) above.

Reviewer 3 Report

General Comments:

 The manuscript assessed the knowledge, attitude, and practices of tomato growing households using questionnaire data captured broad from selected tomato producing regions in Kenya and Uganda, in regarding to T. absoluta, and determination of the potential demand for IPM strategy. The research obtained very interesting and valuable results, shaped the policy direction on scaling up the alternative methods for management of the invasive pest. While it need minor revision.

Specific Comments:

Line 83-94 Pls. convert to a description of the research objectives.

Line 95-105 Pls. do not need to show results of investigation in Introduction.

Line 104-107, Pls. can remove it. But can introduce briefly the research ideas and schemes before the objectives.

Author Response

Response to Reviewer 3 Comments for paper

“Knowledge, Attitude, and Practices on Tomato Leaf 3 Miner, Tuta absoluta on Tomato and Potential Demand 4 for Integrated Pest Management among Smallholder 5 Farmers in Kenya and Uganda”

Thank you for taking the time to read our paper and for the useful feedback that has improved our paper. Below we list your comment and our responses. We hope you will find our responses acceptable.

General comments

Point 1: English language and style are fine/minor spell check required

Response 1: Thank you for the positive response and the suggestions. We have thoroughly read through the paper for spell check.

Point 2: The manuscript assessed the knowledge, attitude, and practices of tomato growing households using questionnaire data captured broad from selected tomato-producing regions in Kenya and Uganda, in regarding to T. absoluta, and determination of the potential demand for IPM strategy. The research obtained very interesting and valuable results, shaped the policy direction on scaling up the alternative methods for management of the invasive pest. While it needs minor revision.

Response 2: Thank you for the positive review of our paper. We have thoroughly read through the paper and made the suggested changes. See specific responses to your comments below.

Specific comments

Point 3: Line 83-94 Pls. convert to a description of the research objectives.

Response 3: Thank you for the positive comment. We have converted it into a description of research objectives.

Point 4: Comment:  Line 95-105 Pls. do not need to show results of investigation in Introduction.

Response 4: Thank you for this observation.  We have reviewed a few papers published in this journal that similarly follow this order of the introduction, and are also further motivated by Evans, 2020. We would therefore wish to retain the results in the introduction so that we can inform our readers on what to expect later in the main result section based on our study objectives.

Point 5: Line 104-107, Pls. can remove it. But can introduce briefly the research ideas and schemes before the objectives

Response 5: Thank you for the positive response on the paper. We have reviewed the paper and edited it out.

Reference:

  1. Evans D. (2020): How to Write the Introduction of Your Development Economics Paper. https://www.cgdev.org/blog/how-write-introduction-your-development-economics-paper
